# Deep Variational Implicit Processes

**Luis A. Ortega[1], Simón Rodríguez Santana[2], Daniel Hernández-Lobato[1]**
[1]Universidad Autónoma de Madrid  [2]ICMAT-CSIC
{luis.ortega,daniel.hernandez}@uam.es, simon.rodriguez@icmat.es

## Abstract

Implicit processes (IPs) are a generalization of Gaussian processes (GPs). IPs may lack a closed-form expression but are easy to sample from. Examples include, among others, Bayesian neural networks or neural samplers. IPs can be used as priors over functions, resulting in flexible models with well-calibrated prediction uncertainty estimates. Methods based on IPs usually carry out function-space approximate inference, which overcomes some of the difficulties of parameter-space approximate inference. Nevertheless, the approximations employed often limit the expressiveness of the final model, resulting, *e.g.*, in a Gaussian predictive distribution, which can be restrictive. We propose here a multi-layer generalization of IPs called the Deep Variational Implicit process (DVIP). This generalization is similar to that of deep GPs over GPs, but it is more flexible due to the use of IPs as the prior distribution over the latent functions. We describe a scalable variational inference algorithm for training DVIP and show that it outperforms previous IP-based methods and also deep GPs. We support these claims via extensive regression and classification experiments. We also evaluate DVIP on large datasets with up to several million data instances to illustrate its good scalability and performance.

## 1 Introduction

The Bayesian approach has become popular for capturing the uncertainty associated to the predictions made by models that otherwise provide point-wise estimates, such as neural networks (NNs) (Gelman et al., 2013; Gal, 2016; Murphy, 2012). However, when carrying out Bayesian inference, obtaining the posterior distribution in the space of parameters can become a limiting factor since it is often intractable. Symmetries and strong dependencies between parameters make the approximate inference problem much more complex. This is precisely the case in large deep NNs. Nevertheless, all these issues can be alleviated by carrying out approximate inference in the space of functions, which presents certain advantages due to the simplified problem. This makes the approximations obtained in this space more precise than those obtained in parameter-space, as shown in the literature (Ma et al., 2019; Sun et al., 2019; Rodríguez Santana et al., 2022; Ma and Hernández-Lobato, 2021).

A recent method for function-space approximate inference is the *Variational Implicit Process* (VIP) (Ma et al., 2019). VIP considers an implicit process (IP) as the prior distribution over the target function. IPs constitute a very flexible family of priors over functions that generalize Gaussian processes (Ma et al., 2019). Specifically, IPs are processes that may lack a closed-form expression, but that are easy-to-sample-from. Examples include Bayesian neural networks (BNN), neural samplers and warped GPs, among others (Rodríguez Santana et al., 2022). Figure 1 (left) shows a BNN, which is a particular case of an IP. Nevertheless, the posterior process of an IP is is intractable most of the times (except in the particular case of GPs). VIP addresses this issue by approximating the posterior using the posterior of a GP with the same mean and covariances as the prior IP. Thus, the approximation used in VIP results in a Gaussian predictive distribution, which may be too restrictive.

Recently, the concatenation of random processes has been used to produce models of increased flexibility. An example are *deep GPs* (DGPs) in which a GP is used as the input of another GP, systematically (Damianou and Lawrence, 2013). Based on the success of DGPs, it is natural to consider the concatenation of IPs to extend their capabilities in a similar fashion to DGPs. Therefore, we introduce in this paper deep VIPs (DVIPs), a multi-layer extension of VIP that provides increased expressive power, enables more accurate predictions, gives better calibrated uncertainty estimates, and captures more complex patterns in the data. Figure 1 (right) shows the architecture considered in

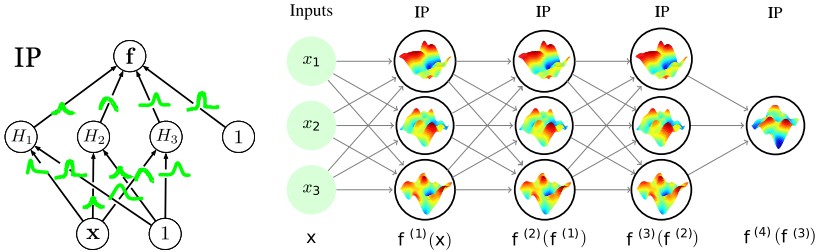

Figure 1: (left) IP resulting from a BNN with random weights and biases following a Gaussian distribution. A sample of the weights and biases generates a random function. (right) Deep VIP in which the input to an IP is the output of a previous IP. We consider a fully connected architecture.

DVIP. Each layer contains several IPs that are approximated using VIP. Importantly, the flexibility of the IP-based prior formulation enables numerous models as the prior over functions, leveraging the benefits of, *e.g.*, convolutional NNs, that increase the performance on image datasets. Critically, DVIP can adapt the prior IPs to the observed data, resulting in improved performance. When GP priors are considered, DVIP is equivalent to a DGP. Thus, it can be seen as a generalization of DGPs.

Approximate inference in DVIPs is done via variational inference (VI). We achieve computational scalability in each unit using a linear approximation of the GP that approximates the prior IP, as in VIP (Ma et al., 2019). The predictive distribution of a VIP is Gaussian. However, since the inputs in the second and following layers are random in DVIP, the final predictive distribution is non-Gaussian. This predictive distribution is intractable. Nevertheless, one can easily sample from it by propagating samples through the IP network shown in Figure 1 (right). This also enables a Monte Carlo approximation of the VI objective which can be optimized using stochastic techniques, as in DGPs (Salimbeni and Deisenroth, 2017). Generating the required samples is straightforward given that the variational posterior depends only on the output of the the previous layers. This results in an iterative sampling procedure that can be conducted in an scalable manner. Importantly, the direct evaluation of covariances are not needed in DVIP, further reducing its cost compared to that of DGPs. The predictive distribution is a mixture of Gaussians (non-Gaussian), more flexible than that of VIP.

We evaluate DVIP in several experiments, both in regression and classification. They show that DVIP outperforms a single-layer VIP with a more complex IP prior. DVIP is also faster to train. We also show that DVIP gives results similar and often better than those of DGPs (Salimbeni and Deisenroth, 2017), while having a lower cost and improved flexibility (due to the more general IP prior). Our experiments also show that adding more layers in DVIP does not over-fit and often improves results.

## 2 BACKGROUND

We introduce the needed background on IPs and the posterior approximation based on a linear model that will be used later on. First, consider the problem of inferring an unknown function $f : \mathbb{R}^M \to \mathbb{R}$ given noisy observations $\mathbf{y} = (y_1, \ldots, y_N)^\mathrm{T}$ at $\mathbf{X} = (\mathbf{x}_1, \ldots, \mathbf{x}_N)$. In the context of Bayesian inference, these observations are related to $\mathbf{f} = (f(\mathbf{x}_1), \ldots, f(\mathbf{x}_N))^\mathrm{T}$ via a likelihood, denoted as $p(\mathbf{y}|\mathbf{f})$. IPs represent one of many ways to define a distribution over a function (Ma et al., 2019).

**Definition 1.** *An IP is a collection of random variables $f(\cdot)$ such that any finite collection $\mathbf{f} = \{f(\mathbf{x}_1), f(\mathbf{x}_2), \ldots, f(\mathbf{x}_N)\}$ is implicitly defined by the following generative process*

$$\mathbf{z} \sim p_{\mathbf{z}}(\mathbf{z}) \quad and \quad f(\mathbf{x}_n) = g_{\boldsymbol{\theta}}(\mathbf{x}_n, \mathbf{z}), \quad \forall n = 1, \ldots, N. \tag{1}$$

An IP is denoted as $f(\cdot) \sim \mathcal{IP}(g_{\boldsymbol{\theta}}(\cdot, \cdot), p_{\mathbf{z}})$, with $\boldsymbol{\theta}$ its parameters, $p_{\mathbf{z}}$ a source of noise, and $g_{\boldsymbol{\theta}}(\mathbf{x}_n, \mathbf{z})$ a function that given $\mathbf{z}$ and $\mathbf{x}_n$ outputs $f(\mathbf{x}_n)$. $g_{\boldsymbol{\theta}}(\mathbf{x}_n, \mathbf{z})$ can be, *e.g.*, a NN with weights specified by $\mathbf{z}$ and $\boldsymbol{\theta}$ using the reparametrization trick (Kingma and Welling, 2014). See Figure 1 (left). Given $\mathbf{z} \sim p_{\mathbf{z}}(\mathbf{z})$ and $\mathbf{x}$, it is straight-forward to generate a sample $f(\mathbf{x})$ using $g_{\boldsymbol{\theta}}$, *i.e.*, $f(\mathbf{x}) = g_{\boldsymbol{\theta}}(\mathbf{x}, \mathbf{z})$.

Consider an IP as the prior for an unknown function and a suitable likelihood $p(\mathbf{y}|\mathbf{f})$. In this context, both the prior $p(\mathbf{f}|\mathbf{X})$ and the posterior $p(\mathbf{f}|\mathbf{X}, \mathbf{y})$ are generally intractable, since the IP assumption does not allow for point-wise density estimation, except in the case of a GP. To overcome this, in Ma

et al. (2019) the model's joint distribution, $p(\mathbf{y}, \mathbf{f}|\mathbf{X})$, is approximated as

$$p(\mathbf{y}, \mathbf{f}|\mathbf{X}) \approx q(\mathbf{y}, \mathbf{f}|\mathbf{X}) = p(\mathbf{y}|\mathbf{f})q_{\mathcal{GP}}(\mathbf{f}|\mathbf{X}), \tag{2}$$

where $q_{\mathcal{GP}}$ is a Gaussian process with mean and covariance functions $m(\cdot)$ and $\mathcal{K}(\cdot, \cdot)$, respectively. These two functions are in turn defined by the mean and covariance functions of the prior IP,

$$m(\mathbf{x}) = \mathbb{E}[f(\mathbf{x})], \qquad \mathcal{K}(\mathbf{x}_1, \mathbf{x}_2) = \mathbb{E}\left[(f(\mathbf{x}_1) - m(\mathbf{x}_1))(f(\mathbf{x}_2) - m(\mathbf{x}_2))\right], \tag{3}$$

which can be estimated empirically by sampling from $\mathrm{IP}(g_{\boldsymbol{\theta}}(\cdot, \cdot), p_{\mathbf{z}})$ (Ma et al., 2019). Using $S$ Monte Carlo samples $f_s(\cdot) \sim \mathrm{IP}(g_{\boldsymbol{\theta}}(\cdot, \cdot), p_{\mathbf{z}})$, $s \in 1, \ldots, S$, the mean and covariance functions are

$$m^\star(\mathbf{x}) = \frac{1}{S}\sum_{s=1}^S f_s(\mathbf{x}), \qquad \mathcal{K}^\star(\mathbf{x}_1, \mathbf{x}_2) = \phi(\mathbf{x}_1)^\mathrm{T}\phi(\mathbf{x}_2),$$
$$\phi(\mathbf{x}_n) = 1/\sqrt{S}\left(f_1(\mathbf{x}_n) - m^\star(\mathbf{x}_n), \ldots, f_S(\mathbf{x}_n) - m^\star(\mathbf{x}_n)\right)^\mathrm{T}. \tag{4}$$

Thus, the VIP's prior for $f$ is simply a GP approximating the prior IP, which can be, *e.g.*, a BNN. Critically, the samples $f_s(\mathbf{x})$ keep the dependence w.r.t. the IP prior parameters $\boldsymbol{\theta}$, which enables prior adaptation to the observed data in VIP (Ma et al., 2019). Unfortunately, this formulation has the typical cost in $\mathcal{O}(N^3)$ of GPs (Rasmussen and Williams, 2006). To solve this and also allow for mini-batch training, the GP is approximated using a linear model: $f(\mathbf{x}) = \phi(\mathbf{x})^\mathrm{T}\mathbf{a} + m^\star(\mathbf{x})$, where $\mathbf{a} \sim \mathcal{N}(\mathbf{0}, \mathbf{I})$. Under this definition, the prior mean and covariances of $f(\mathbf{x})$ are given by (4). The posterior of $\mathbf{a}$, $p(\mathbf{a}|\mathbf{y})$, is approximated using a Gaussian distribution, $q_{\boldsymbol{\omega}}(\mathbf{a}) = \mathcal{N}(\mathbf{a}|\mathbf{m}, \mathbf{S})$, whose parameters $\boldsymbol{\omega} = \{\mathbf{m}, \mathbf{S}\}$ (and other model's parameters) are adjusted by maximizing the $\alpha$-energy

$$\mathcal{L}^\alpha(\boldsymbol{\omega}, \boldsymbol{\theta}, \sigma^2) = \sum_{n=1}^N \frac{1}{\alpha}\log \mathbb{E}_{q_\omega}\left[\mathcal{N}(y_n|f(\mathbf{x}_n), \sigma^2)^\alpha\right] - \mathrm{KL}\left(q_\omega(\mathbf{a})|p(\mathbf{a})\right), \tag{5}$$

where $\alpha = 0.5$, a value that provides good general results (Hernández-Lobato et al., 2016; Rodríguez Santana and Hernández-Lobato, 2022). A Gaussian likelihood is also assumed with variance $\sigma^2$. Otherwise, 1-dimensional quadrature methods are required to evaluate (5). Importantly, (5) allows for mini-batch training to estimate $\boldsymbol{\omega}$, $\boldsymbol{\theta}$ and $\sigma^2$ from the data. Given, $q_{\boldsymbol{\omega}}(\mathbf{a})$, it is straight-forward to make predictions. The predictive distribution for $y_n$ is, however, limited to be Gaussian in regression.

## 3 DEEP VARIATIONAL IMPLICIT PROCESSES

Deep variational implicit processes (DVIPs) are models that consider a deep implicit process as the prior for the latent function. A deep implicit process is a concatenation of multiple IPs, recursively defining an implicit prior over latent functions. Figure 1 (right) illustrates the architecture considered, which is fully connected. The prior on the function at layer $l$ and unit $h$, $f_h^l(\cdot)$, is an independent IP whose inputs are given by the outputs of the previous layer. Let $H_l$ be the $l$-th layer dimensionality.

**Definition 2.** *A deep implicit process is a collection of random variables $\{f_{h,n}^l : l = 1, \ldots, L \wedge h = 1 \ldots, H_l \wedge n = 1, \ldots, N\}$ such that each $f_{h,n}^l = f_h^l(\mathbf{f}_{\cdot,n}^{l-1})$, with $\mathbf{f}_{\cdot,n}^{l-1}$ the output of the previous layer in the network, i.e., $\mathbf{f}_{\cdot,n}^{l-1} = (f_{1,n}^{l-1}, \ldots, f_{H_{l-1},n}^{l-1})^T$, and each $f_h^l(\cdot)$ an independent IP: $f_h^l(\cdot) \sim \mathcal{IP}(g_{\boldsymbol{\theta}_h^l}(\cdot, \mathbf{z}), p_{\mathbf{z}})$, where $\mathbf{f}_{\cdot,n}^0 = \mathbf{x}_n$ symbolizes the initial input features to the network.*

As in VIP, we consider GP approximations for all the IPs in the deep IP prior defined above. These GPs are further approximated using a linear model, as in VIP. This provides an expression for $f_{h,n}^l$ given the previous layer's output $\mathbf{f}_{\cdot,n}^{l-1}$ and $\mathbf{a}_h^l$, the coefficients of the linear model for the unit $h$ at layer $l$. Namely, $f_{h,n}^l = \phi_h^l(\mathbf{f}_{\cdot,n}^{l-1})^\mathrm{T}\mathbf{a}_h^l + m_{h,l}^\star(\mathbf{f}_{\cdot,n}^{l-1})$, where $\phi_h^l(\cdot)$ and $m_{h,l}^\star(\cdot)$ depend on the prior IP parameters $\boldsymbol{\theta}_h^l$. To increase the flexibility of the model, we consider latent Gaussian noise around each $f_{h,n}^l$ with variance $\sigma_{l,h}^2$ (except for the last layer $l = L$). That is, $\boldsymbol{\sigma}_l^2 = \{\sigma_{l,h}^2\}_{h=1}^{H_l}$ are the noise variances at layer $l$. Then, $p(f_{h,n}^l|\mathbf{f}_{\cdot,n}^{l-1}, \mathbf{a}_h^l)$ is a Gaussian distribution with mean $\phi_h^l(\mathbf{f}_{\cdot,n}^{l-1})^\mathrm{T}\mathbf{a}_h^l + m_{h,l}^\star(\mathbf{f}_{\cdot,n}^{l-1})$ and variance $\sigma_{l,h}^2$. Let $\mathbf{A}^l = \{\mathbf{a}_1^l, \ldots, \mathbf{a}_{H_l}^l\}$ and $\mathbf{F}^l = \{\mathbf{f}_{\cdot,1}^l, \ldots, \mathbf{f}_{\cdot,N}^l\}$. The joint distribution of all the variables (observed and latent) in DVIP is

$$p\left(\mathbf{y}, \{\mathbf{F}^l, \mathbf{A}^l\}_{l=1}^L\right) = \prod_{n=1}^N p(y_n|\mathbf{f}_{\cdot,n}^L) \prod_{l=1}^L \prod_{h=1}^{H_l} p(f_{h,n}^l|\mathbf{a}_h^l)p(\mathbf{a}_h^l), \tag{6}$$

where $p(\mathbf{a}_h^l) = \mathcal{N}(\mathbf{a}_h^l|\mathbf{0}, \mathbf{I})$ and we have omitted the dependence of $f_{h,n}^l$ on $\mathbf{f}_n^{l-1}$ to improve readability. In (6), $\prod_{n=1}^N p(y_n|\mathbf{f}_{\cdot,n}^L)$ is the likelihood and $\prod_{n=1}^N \prod_{l=1}^L \prod_{h=1}^{h_l} p(f_{h,n}^l|\mathbf{a}_h^l)p(\mathbf{a}_h^l)$ is the deep IP prior. It may seem that the prior assumes independence across points. Dependencies are, however, obtained by marginalizing out each $\mathbf{a}_h^l$, which is tractable since the model is linear in $\mathbf{a}_h^l$.

We approximate the posterior $p\left(\{\mathbf{F}^l, \mathbf{A}^l\}_{l=1}^L | \mathbf{y}\right)$ using an approximation with a fixed and a tunable part, simplifying dependencies among layer units, but maintaining dependencies between layers:

$$q(\{\mathbf{F}^l, \mathbf{A}^l\}_{l=1}^L) = \prod_{n=1}^N \prod_{l=1}^L \prod_{h=1}^{H_l} p(f_{h,n}^l | \mathbf{a}_h^l) q(\mathbf{a}_h^l), \qquad q(\mathbf{a}_h^l) = \mathcal{N}(\mathbf{a}_h^l | \mathbf{m}_h^l, \mathbf{S}_h^l), \quad (7)$$

where the factors $p(f_{h,n}^l | \mathbf{a}_h^l)$ are fixed to be the same factors as those in (6), and the factors $q(\mathbf{a}_d^l)$ are the ones being specifically tuned. This approximation resembles that of Salimbeni and Deisenroth (2017) for DGPs, since the conditional distribution is fixed. However, we do not employ sparse GPs based on inducing points and use a linear model to approximate the posterior at each layer instead. Computing $q(\mathbf{f}_{\cdot,n}^L)$ is intractable. However, one can easily sample from it, as described next.

Using (7), we can derive a variational lower bound at whose maximum the Kullback-Leibler (KL) divergence between $q(\{\mathbf{F}^l, \mathbf{A}^l\}_{l=1}^L)$ and $p\left(\{\mathbf{F}^l, \mathbf{A}^l\}_{l=1}^L | \mathbf{y}\right)$ is minimized. Namely,

$$\mathcal{L}\left(\Omega, \Theta, \{\boldsymbol{\sigma}_l^2\}_{l=1}^{L-1}\right) = \sum_{n=1}^N \mathbb{E}_q\left[\log p\left(y_n | \mathbf{f}_{\cdot,n}^L\right)\right] - \sum_{l=1}^L \sum_{h=1}^{H_l} \mathrm{KL}\left(q(\mathbf{a}_h^l) | p(\mathbf{a}_h^l)\right), \quad (8)$$

where we have used the cancellation of factors, and where $\Omega = \{\mathbf{m}_h^l, \mathbf{S}_h^l : l = 1, \dots, L \wedge h = 1, \dots, H_l\}$ are the parameters of $q$ and $\Theta = \{\boldsymbol{\theta}_h^l : l = 1, \dots, L \wedge h = 1, \dots, H_l\}$ are the DVIP prior parameters. Furthermore, $\mathrm{KL}(\cdot|\cdot)$ denotes the KL-divergence between distributions. $\mathrm{KL}(q(\mathbf{a}_h^l)|p(\mathbf{a}_h^l))$ involves Gaussian distributions and can be evaluated analytically. The expectations $\mathbb{E}_q\left[\log p(y_n | \mathbf{f}_{\cdot,n}^L)\right]$ are intractable. However, they can be approximated via Monte Carlo, using the reparametrization trick (Kingma and Welling, 2014). Moreover, $\sum_{n=1}^N \mathbb{E}_q\left[\log p(y_n | \mathbf{f}_{\cdot,n}^L)\right]$ can be approximated using mini-batches. Thus, (8) can be maximized w.r.t. $\Omega$, $\Theta$ and $\{\boldsymbol{\sigma}_l^2\}_{l=1}^{L-1}$, using stochastic optimization. Maximization w.r.t. $\Theta$ allows for prior adaptation to the observed data, which is a key factor when considering IP priors. Appendix A has all the details about the derivation of (8).

**Sampling from the marginal posterior approximation.** The evaluation of (8) requires samples from $q(\mathbf{f}_{\cdot,n}^L)$ for all the instances in a mini-batch. This marginal only depends on the variables of the inner layers and units $f_{h,n}^l$ corresponding to the $n$-th instance. See Appendix B. Thus, we can sample from $q(\mathbf{f}_{\cdot,n}^L)$ by recursively propagating samples from the first to the last layer, using $\mathbf{x}_n$ as the input. Specifically, $p(f_{h,n}^l | \mathbf{f}_{\cdot,n}^{l-1}, \mathbf{a}_h^l)$ is Gaussian with a linear mean in terms of $\mathbf{a}_h^l$, and $q(\mathbf{a}_h^l)$ is Gaussian. Thus, $q(f_{h,n}^l | \mathbf{f}_{\cdot,n}^{l-1}) = \int p(f_{h,n}^l | \mathbf{f}_{\cdot,n}^{l-1}, \mathbf{a}_h^l) q(\mathbf{a}_h^l) \, d\mathbf{a}_h^l$ is also Gaussian with mean and variance:

$$\hat{m}_{h,n}^l(\mathbf{f}_{\cdot,n}^{l-1}) = \boldsymbol{\phi}_h^l(\mathbf{f}_{\cdot,n}^{l-1})^{\mathrm{T}} \mathbf{m}_h^l + m_{h,l}^{\star}(\mathbf{f}_{\cdot,n}^{l-1}), \quad \hat{v}_{h,n}^l(\mathbf{f}_{\cdot,n}^{l-1}) = \boldsymbol{\phi}_h^l(\mathbf{f}_{\cdot,n}^{l-1})^{\mathrm{T}} \mathbf{S}_h^l \boldsymbol{\phi}_h^l(\mathbf{f}_{\cdot,n}^{l-1}) + \sigma_{l,h}^2, \quad (9)$$

where $\sigma_{L,h}^2 = 0$ and $\mathbf{m}_h^l$ and $\mathbf{S}_h^l$ are the parameters of $q(\mathbf{a}_h^l)$. Initially, let $l = 1$. We set $\hat{\mathbf{f}}_{\cdot,n}^0 = \mathbf{x}_n$ and generate a sample from $q(f_{h,n}^l | \hat{\mathbf{f}}_{\cdot,n}^0)$ for $h = 1, \dots, H_l$. Let $\hat{\mathbf{f}}_{\cdot,n}^l$ be that sample. Then, we use $\hat{\mathbf{f}}_{\cdot,n}^l$ as the input for the next layer. This process repeats for $l = 2, \dots, L$, until we obtain $\hat{\mathbf{f}}_{\cdot,n}^L \sim q(\mathbf{f}_{\cdot,n}^L)$.

**Making predictions for new instances.** Let $\mathbf{f}_{\cdot,\star}^L$ be the values at the last layer for the new instance $\mathbf{x}_\star$. We approximate $q(\mathbf{f}_{\cdot,\star}^L)$ by propagating $R$ Monte Carlo samples through the network. Then,

$$q(\mathbf{f}_{\cdot,\star}^L) \approx R^{-1} \sum_{r=1}^R \prod_{h=1}^{H_L} \mathcal{N}(f_{h,\star}^L | \hat{m}_{h,\star}^L(\hat{\mathbf{f}}_{\cdot,\star}^{L-1,r}), \hat{v}_{h,\star}^L(\hat{\mathbf{f}}_{\cdot,\star}^{L-1,r})), \quad (10)$$

where $\hat{\mathbf{f}}_{\cdot,\star}^{L-1,r}$ is the $r$-th sample arriving at layer $L-1$ and $\hat{m}_{h,\star}^L(\cdot)$ and $\hat{v}_{h,\star}^L(\cdot)$ are given by (9). We note that (10) is a Gaussian mixture, which is expected to be more flexible than the Gaussian predictive distribution of VIP. We set $R$ to 100 for testing and to 1 for training, respectively. Computing, $p(y_\star) = \mathbb{E}_q[p(y_\star | \mathbf{f}_{\cdot,\star}^L)]$ is tractable in regression, and can be approximated using 1-dimensional quadrature in binary and multi-class classification, as in DGPs (Salimbeni and Deisenroth, 2017).

**Input propagation.** Inspired by the *skip layer* approach of, *e.g.* ResNet (He et al., 2016), and the addition of the original input to each layer in DGPs (Duvenaud et al., 2014; Salimbeni and Deisenroth, 2017), we implement the same approach here. For this, we add the previous input to the mean in (9) if the input and output dimension of the layer is the same, except in the last layer, where the added mean is zero. Namely, $\hat{m}_{h,n}^l(\mathbf{f}_{\cdot,n}^{l-1}) = \boldsymbol{\phi}_h^l(\mathbf{f}_{\cdot,n}^{l-1})^{\mathrm{T}} \mathbf{m}_h^l + m_{h,l}^{\star}(\mathbf{f}_{\cdot,n}^{l-1}) + f_{h,n}^{l-1}$, for $l = 1, \dots, L-1$.

**Computational cost.** The cost at layer $l$ in DVIP is in $\mathcal{O}(BS^2 H_l)$, if $B > S$, with $S$ the number of samples from the prior IPs, $B$ the size of the mini-batch and $H_l$ the output dimension of the layer. This cost is similar to that of a DGP, which has a squared cost in terms of $M$, the number of inducing

points (Hensman et al., 2013; Bui et al., 2016; Salimbeni and Deisenroth, 2017). In Salimbeni and Deisenroth (2017), the cost at each layer is $\mathcal{O}(BM^2 H_l)$, if $B > M$. In our work, however, the number of prior IP samples $S$ is smaller than the typical number of inducing points in DGPs. In our experiments we use $S = 20$, as suggested for VIP (Ma et al., 2019). Considering a DVIP with $L$ layers, the total cost is $\mathcal{O}(BS^2(H_1 + \cdots + H_L))$. Our experiments show that, despite the generation of the prior IP samples, DVIP is faster than DGP, and the gap becomes bigger as $L$ increases.

## 4 RELATED WORK

The relation between GPs and IPs has been previously studied. A 1-layer BNN with cosine activations and infinite width is equivalent to a GP with RBF kernel (Hensman et al., 2017). A deep BNN is equivalent to a GP with a compositional kernel (Cho and Saul, 2009), as shown by Lee et al. (2017). These methods make possible to create expressive kernels for GPs. An inverse reasoning is used by Flam-Shepherd et al. (2017), where GP prior properties are encoded into the prior weights of a BNN. Fortuin (2022) provides an extensive review about prior distributions in function-space defined by BNNs and stochastic processes. They give methods of learning priors for these models from data.

VIP (Ma et al., 2019) arises from the treatment of BNNs as instances of IPs. For this, an approximate GP is used to assist inference. Specifically, a prior GP is built with mean and covariance function given by the prior IP, a BNN. VIP can make use of the more flexible IP prior, whose parameters can be inferred from the data, improving results over GPs (Ma et al., 2019). However, VIP's predictive distribution is Gaussian. DVIP overcomes this problem providing a non-Gaussian predictive distribution. Thus, it is expected to lead to a more flexible model with better calibrated uncertainty estimates. DVIP differs from deep kernel learning (DKL) (Wilson et al., 2016a;b), where a GP is applied to a non-linear transformation of the inputs. Its predictive distribution is Gaussian, unlike that of DVIP and the non-linear transformation of DKL ignores epistemic uncertainty, unlike IPs.

There are other methods that have tried to make inference using IPs. Sun et al. (2019) propose the *functional Bayesian neural networks* (fBNN), where a second IP is used to approximate the posterior of the first IP. This is a more flexible approximation than that of VIP. However, because both the prior and the posterior are implicit, the noisy gradient of the variational ELBO is intractable and has to be approximated. For this, a spectral gradient estimator is used (Shi et al., 2018). To ensure that the posterior IP resembles the prior IP in data-free regions, fBNN relies on uniformly covering the input space. In high-dimensional spaces this can lead to poor results. Moreover, because of the spectral gradient estimator fBNN cannot tune the prior IP parameters to the data. In the particular case of a GP prior, fBNN simply maximizes the marginal likelihood of the GP w.r.t. the prior parameters. However, a GP prior implies a GP posterior. This questions using a second IP for posterior approximation. Recent works have employed a first order Taylor approximation to linearize the IP, approximating their implicit distribution by a GP (Rudner et al., 2021; Immer et al., 2021). This leads to another GP approximation to an IP, different of VIP, where the computational bottleneck is located at computing the Jacobian of the linearized transformation instead of samples from the prior. More recent work in parameter-space considers using a repulsive term in BNN ensembles to guarantee the diversity among the members, avoiding their collapse in the parameter space (D'Angelo and Fortuin, 2021).

Sparse implicit processes (SIPs) use inducing points for approximate inference in the context of IPs (Rodríguez Santana et al., 2022). SIP does not have the limitations of neither VIP nor fBNN. It produces flexible predictive distributions (Gaussian mixtures) and it can adjust its prior parameters to the data. SIP, however, relies on a classifier to estimate the KL-term in the variational ELBO, which adds computational cost. SIP's improvements over VIP are orthogonal to those of DVIP over VIP and, in principle, SIP may also be used as the building blocks of DVIP, leading to even better results.

Functional variational inference (FVI) minimizes the KL-divergence between stochastic process for approximate inference (Ma and Hernández-Lobato, 2021). Specifically, between the model's IP posterior and a second IP, as in fBNN. This is done efficiently by approximating first the IP prior using a stochastic process generator (SPG). Then, a second SPG is used to efficiently approximate the posterior of the previous SPG. Both SPGs share key features that make this task easy. However, FVI is also limited, as fBNN, since it cannot adjust the prior to the data. This questions its practical utility.

As shown by Rodríguez Santana et al. (2022), adjusting the prior IP to the observed data is key for accurate predictions. This discourages using fBNN and FVI as building blocks of a model using deep

IP priors on the target function. Moreover, these methods do not consider deep architectures such as the one in Figure 1 (right). Therefore, we focus on comparing with VIP, as DVIP generalizes VIP.

To our knowledge, the concatenation of IPs with the goal of describing priors over functions has not been studied previously. However, the concatenation of GPs resulting in deep GPs (DGPs), has received a lot of attention (Lawrence and Moore, 2007; Bui et al., 2016; Cutajar et al., 2017; Salimbeni and Deisenroth, 2017; Havasi et al., 2018; Yu et al., 2019). In principle, DVIP is a generalization of DGPs in the same way as IPs generalize GPs. Namely, the IP prior of each layer's unit can simply be a GP. Samples from such a GP prior can be efficiently obtained using, *e.g.*, a 1-layer BNN with cosine activation functions that is wide enough (Rahimi and Recht, 2007; Cutajar et al., 2017).

The posterior approximation used in DVIP is similar to that used in the context of DGPs by Salimbeni and Deisenroth (2017). Moreover, each prior IP is approximated by a GP in DVIP. However, in spite of these similarities, there are important differences between our work and that of Salimbeni and Deisenroth (2017). Specifically, instead of relying on sparse GPs to approximate each GP within the GP network, DVIP uses a linear GP approximation that needs no specification of inducing points. Furthermore, the covariance function used in DVIP is more flexible and specified by the assumed IP prior. DGPs are, on the other hand, restricted to GP priors with specific covariance functions. DVIP has the extra flexibility of considering a wider range of IP priors that need not be GPs. Critically, in our experiments, DVIP significantly outperforms DGPs in image-related datasets, where using specific IP priors based, *e.g.*, on convolutional neural networks, can give a significant advantage over standard DGPs. Our experiments also show that DVIP is faster to train than the DGP of Salimbeni and Deisenroth (2017), and the difference becomes larger as the number of layers $L$ increases.

## 5 EXPERIMENTS

We evaluate the proposed method, DVIP, on several tasks. We use $S = 20$ and a BNN as the IP prior for each unit. These BNNs have 2 layers of 10 units each with *tanh* activations, as in Ma et al. (2019). We compare DVIP with VIP (Ma et al., 2019) and DGPs, closely following Salimbeni and Deisenroth (2017). We do not compare results with fBNN nor FVI, described in Section 4, because they cannot tune the prior IP parameters to the data nor they do consider deep architectures as the one in Figure 1 (right). An efficient PyTorch implementation of DVIP is found in the supplementary material. Appendix C has all the details about the experimental settings considered for each method.

**Regression UCI benchmarks.** We compare each method on 8 regression datasets from the UCI Repository (Dua and Graff, 2017). Following common practice, we validate the performance using 20 different train / test splits of the data with $10\%$ test size (Hernández-Lobato and Adams, 2015). We evaluate DVIP and DGP using 2, 3, 4 and 5 layers. We compare results with VIP, which is equivalent to DVIP with $L = 1$, and with VIP using a bigger BNN of 200 units per layer. We also compare results with a single sparse GP, which is equivalent to DGP for $L = 1$, and with a Sparse Implicit Processes (SIP) with the same prior (Rodríguez Santana et al., 2022). Figure 2 shows the results obtained in terms of the negative test log-likelihood. Results in terms of the RMSE and the exact figures are found in Appendix H. DVIP with at least 3 layers performs best on 4 out of the 8 datasets (`Boston`, `Energy`, `Concrete` and `Power`), having comparable results on `Winered` and `Naval` (all methods have zero RMSE on this dataset). DGPs perform best on 2 datasets (`Protein` and `Kin8nm`), but the differences are small. Appendix G shows that using a GP prior in DVIP in these problems performs better at a higher cost. Adding more layers in DVIP does not lead to over-fitting and it gives similar and often better results (notably on larger datasets: `Naval`, `Protein`, `Power` and `Kin8nm`). DVIP also performs better than VIP and SIP most of the times. By contrast, using a more flexible BNN prior in VIP (*i.e.*, 200 units) does not improve results. Figure 3 shows the training time in seconds of each method. DVIP is faster than DGP and faster than VIP with the 200 units BNN prior. Summing up, DVIP achieves similar results to those of DGPs, but at a smaller cost.

**Interpolation results.** We carry out experiments on the CO2 time-series dataset (https://scrippsco2.ucsd.edu). This dataset has $CO_2$ measurements from the Mauna Loa Observatory, Hawaii, in 1978. We split the dataset in five consecutive and equal parts, and used the 2nd and 4th parts as test data. All models are trained for $100,000$ iterations. Figure 4 shows the predictive distribution of DVIP and DGP with $L = 2$ on the data. DVIP captures the data trend in the missing gaps. For DVIP we show samples from the learned prior, which are very smooth. By contrast, a

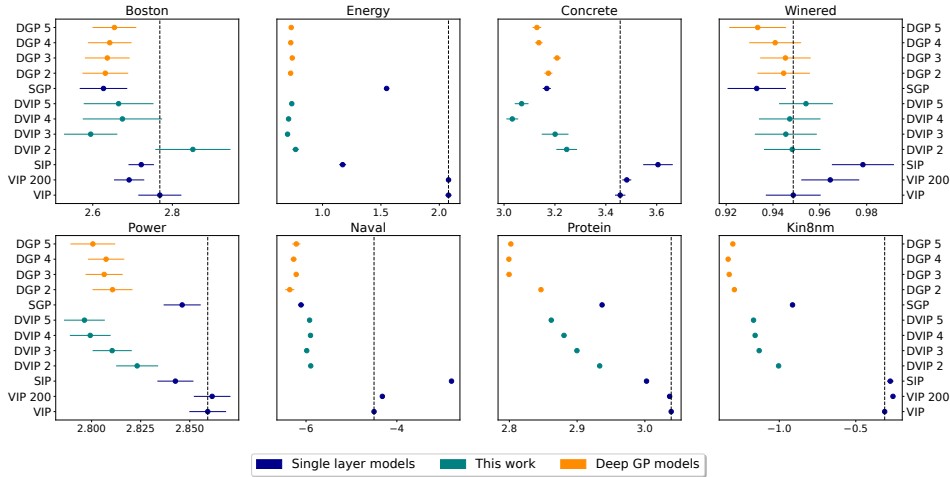

Figure 2: Negative test log-likelihood results on regression UCI benchmark datasets over 20 splits. We show standard errors. Lower values (to the left) are better.

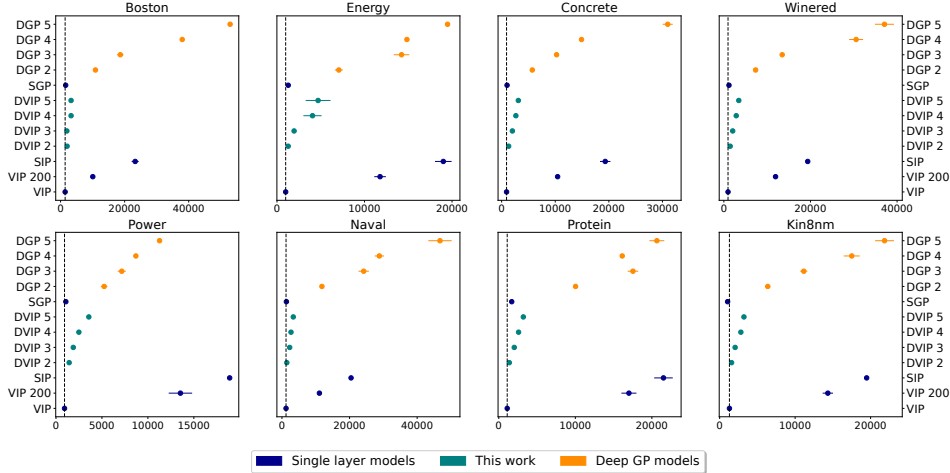

Figure 3: CPU training time (in seconds) on regression UCI benchmark datasets over 20 splits. We show standard errors. Lower values (to the left) are better.

DGP with RBF kernels fails to capture the data trend, leading to mean reversion and over-estimation of the prediction uncertainty (similar results for SGP are shown in Appendix H). Thus, the BNN prior considered by DVIP could be a better choice here. This issue of DGPs can be overcome using compositional kernels (Duvenaud et al., 2014), but that requires using kernel search algorithms.

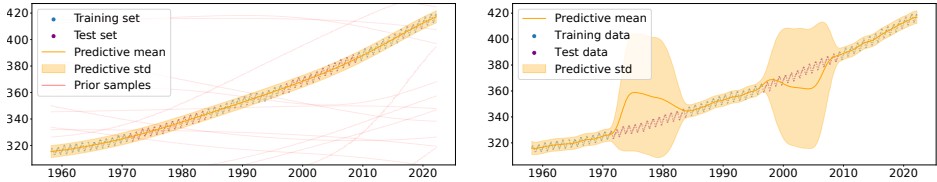

Figure 4: Missing values interpolation results on the CO2 dataset. Predictive distribution of DVIP (left) and DGP (right) with 2 layers each. Two times the standard deviation is represented.

**Large scale regression.** We evaluate each method on 3 large regression datasets. First, the Year dataset (UCI) with $515,345$ instances and $90$ features, where the original train/test splits are used.

Second, the US flight delay (Airline) dataset (Dutordoir et al., 2020; Hensman et al., 2017), where following Salimbeni and Deisenroth (2017) we use the first $700,000$ instances for training and the next $100,000$ for testing. 8 features are considered: *month, day of month, day of week, plane age, air time, distance, arrival time and departure time*. For these two datasets, results are averaged over 10 different random seed initializations. Lastly, we consider data recorded on January, 2015 from the Taxi dataset (Salimbeni and Deisenroth, 2017). In this dataset 10 attributes are considered: *time of day, day of week, day of month, month, pickup latitude, pickup longitude, drop-off longitude, drop-off latitude, trip distance and trip duration*. Trips with a duration lower than 10 seconds and larger than 5 hours are removed as in Salimbeni and Deisenroth (2017), leaving $12,695,289$ instances. Results are averaged over 20 train/test splits with 90% and 10% of the data. Here, we trained each method for $500,000$ iterations. The results obtained are shown in Table 1. The last column shows the best result by DGP, which is achieved for $L = 3$ on each dataset. We observe that DVIP outperforms VIP on all datasets, and on Airline and Taxi, the best method is DVIP. In Taxi a sparse GP and DGPs give similar results, while DVIP improves over VIP. The best method on Year, however, is DGP. The difference between DVIP and DGP is found in the prior (BNN vs. GP), and in the approximate inference algorithm (DGP uses inducing points for scalability and DVIP a linear model). Since DVIP generalizes DGP, DVIP using GP priors should give similar results to those of DGP on Year. Appendix G shows, however, that the differences in Year are not only due to the chosen prior (BNN vs. GP) but also due to the posterior approximation (linear model vs. inducing points). VIP using inducing points and a GP prior gives similar results to those of SGP. The cost of approximately sampling from the GP prior in VIP is, however, too expensive to consider adding extra layers in DVIP.

Table 1: Root mean squared error results on large scale regression datasets.

|  | Single-layer | | Ours | | | | DS-DGP |
|---|---|---|---|---|---|---|---|
|  | SGP | VIP | DVIP 2 | DVIP 3 | DVIP 4 | DVIP 5 | DGP 3 |
| Year | 9.15±0.01 | 10.27±0.01 | 9.61±0.03 | 9.34±0.02 | 9.30±0.03 | 9.27±0.03 | **8.94±0.03** |
| Airline | 38.61±0.05 | 38.90±0.06 | 37.96±0.03 | 37.91±0.05 | 37.83±0.03 | **37.80±0.05** | 37.95±0.04 |
| Taxi | 554.22±0.32 | 554.60±0.19 | 549.28±0.59 | **531.42±1.59** | 547.33±1.03 | 538.94±2.23 | 552.90±0.33 |

**Image classification.** We consider the binary classification dataset Rectangles (Salimbeni and Deisenroth, 2017) and the multi-class dataset MNIST (Deng, 2012). Each dataset has $28 \times 28$ pixels images. The Rectangles dataset has $12,000$ images of a (non-square) rectangle. The task is to determine if the height is larger than the width. Here, we used a probit likelihood in each method. The MNIST dataset has $70,000$ images of handwritten digits. The labels correspond with each digit. Here, we used the robust-max multi-class likelihood in each method (Hernández-Lobato et al., 2011). In Rectangles, $20,000$ iterations are enough to ensure convergence. We employ the provided train-test splits for each dataset. Critically, here we exploit DVIP's capability to use more flexible priors. In the first layer we employ a convolutional NN (CNN) prior with two layers of 4 and 8 channels respectively. No input propagation is used in the first layer. The results obtained are shown in Table 2, averaged over 10 random seed initializations. We report the best obtained results for DGP, which are obtained for $L = 3$. We observe that DVIP obtains much better results than those of DGP and VIP in terms of accuracy. DVIP increases accuracy by 11% on Rectangles compared to DGP, probably as a consequence of the CNN prior considered in the first layer of the network being more suited for image-based datasets. Convolutional DGP can perform better than standard DGP in these tasks, however, the objective here is to highlight that DVIP allows to easily introduce domain-specific prior functions that might not be easily used by standard GPs. Image classification is only an example of this. Other examples may include using recurrent network architectures for sequential data.

Table 2: Results on image classification datasets.

| **MNIST** | Single-layer | | Ours | | DS-DGP | |
|---|---|---|---|---|---|---|
|  | SGP | VIP | DVIP 2 | DVIP 3 | DGP 2 | DGP 3 |
| Accuracy (%) | 96.25±0.04 | 97.99±0.03 | **98.39±0.05** | 98.36±0.04 | 97.75±0.04 | 97.86±0.05 |
| Likelihood | -0.146±0.00 | -0.144±0.00 | -0.074±0.00 | -0.080±0.00 | -0.082±0.00 | **−0.072±0.00** |

| **Rectangles** | Single-layer | | Ours | | | | DS-DGP |
|---|---|---|---|---|---|---|---|
|  | SGP | VIP | DVIP 2 | DVIP 3 | DVIP 4 | DVIP 5 | DGP 3 |
| Accuracy (%) | 72.54±0.14 | 85.63±0.18 | 87.84±0.20 | **88.21±0.12** | 87.43±0.20 | 86.49±0.17 | 75.16±0.16 |
| Likelihood | -0.518±0.00 | -0.348±0.00 | -0.306±0.00 | **−0.295±0.00** | -0.309±0.00 | -0.320±0.00 | -0.470±0.00 |
| AUC | 0.828±0.00 | 0.930±0.00 | 0.950±0.00 | **0.953±0.00** | 0.947±0.00 | 0.939±0.00 | 0.858±0.00 |

**Large scale classification.** We evaluate each method on two massive binary datasets: `SUSY` and `HIGGS`, with $5.5$ million and $10$ million instances, respectively. These datasets contain Monte Carlo physics simulations to detect the presence of the Higgs boson and super-symmetry (Baldi et al., 2014). We use the original train/test splits of the data, and train for $500,000$ iterations. We report the AUC metric for comparison with Baldi et al. (2014); Salimbeni and Deisenroth (2017). Results are shown in Table 3, averaged over 10 different random seed initializations. In the case of DGPs, we report the best results, which correspond to $L = 4$ and $L = 5$, respectively. We observe that DVIP achieves the highest performance on `SUSY` (AUC of $0.8756$) which is comparable to that of DGPs ($0.8751$) and to the best reported results in Baldi et al. (2014). Namely, shallow NNs (NN, $0.875$), deep NN (DNN, $0.876$) and boosted decision trees (BDT, $0.863$). On `HIGGS`, despite seeing an steady improvement over VIP by using additional layers, the performance is worse than that of DGP (AUC $0.8324$). Again, we believe that GPs with an RBF kernel may be a better prior here, and that DVIP using inducing points and a GP prior should give similar results to those of DGP. However, the high computational cost of approximately sampling from the GP prior will make this too expensive.

Table 3: Results on large classification datasets.

| SUSY | Single-layer | | Ours | | | | DS-DGP |
|---|---|---|---|---|---|---|---|
| | SGP | VIP | DVIP 2 | DVIP 3 | DVIP 4 | DVIP 5 | DGP 4 |
| Accuracy (%) | 79.75±0.02 | 78.68±0.02 | 80.11±0.03 | 80.13±0.01 | 80.22±0.01 | **80.24±0.02** | 80.06±0.01 |
| Likelihood | -0.436±0.00 | -0.456±0.00 | -0.429±0.00 | -0.429±0.00 | **−0.427±0.00** | **−0.427±0.00** | -0.432±0.00 |
| AUC | 0.8727±0.00 | 0.8572±0.00 | 0.8742±0.00 | 0.8749±0.00 | 0.8755±0.00 | **0.8756±0.00** | 0.8751±0.00 |
| **HIGGS** | SGP | VIP | DVIP 2 | DVIP 3 | DVIP 4 | DVIP 5 | DGP 5 |
| Accuracy (%) | 69.95±0.03 | 57.42±0.03 | 66.09±0.02 | 69.85±0.02 | 70.43±0.01 | 72.01±0.02 | **74.92±0.01** |
| Likelihood | -0.573±0.00 | -0.672±0.00 | -0.611±0.00 | -0.575±0.00 | -0.565±0.00 | -0.542±0.00 | **−0.501±0.00** |
| AUC | 0.7693±0.00 | 0.6247±0.00 | 0.7196±0.00 | 0.7704±0.00 | 0.7782±0.00 | 0.7962±0.00 | **0.8324±0.00** |

**Impact of the Number of Samples $S$ and the Prior Architecture.** Appendix E investigates the impact of the number of samples $S$ on DVIP's performance. The results show that one can get sometimes even better results in DVIP by increasing $S$ at the cost of larger training times. Appendix F shows that changing the structure of the prior BNN does not heavily affect the results of DVIP.

## 6 DISCUSSION

Deep Variational Implicit Process (DVIP), a model based on the concatenation of implicit processes (IPs), is introduced as a flexible prior over latent functions. DVIP can be used on a variety of regression and classification problems with no need of hand-tuning. Our results show that DVIP outperforms or matches the performance of a single layer VIP and GPs. It also gives similar and sometimes better results than those of deep GPs (DGPs). However, DVIP has less computational cost when using a prior that is easy to sample from. Our experiments have also demonstrated that DVIP is both effective and scalable on a wide range of tasks. DVIP does not seem to over-fit on small datasets by increasing the depth, and on large datasets, extra layers often improve performance. We have also showed that increasing the number of layers is far more effective than increasing the complexity of the prior of a single-layer VIP model. Aside from the added computation time, which is rather minor, we see no drawbacks to the use of DVIP instead of a single-layer VIP, but rather significant benefits.

The use of domain specific priors, such as CNNs in the first layer, has provided outstanding results in image-based datasets compared to other GP methods. This establishes a new use of IPs with not-so-general prior functions. We foresee employing these priors in other domain specific tasks, such as forecasting or data encoding, as an emerging field of study. The prior flexibility also results in a generalization of DGPs. As a matter of fact, DVIP gives similar results to those of DGPs if a GP is considered as the IP prior for each unit. Preliminary experiments in Appendix G confirms this.

Despite the good results, DVIP presents some limitations: first of all, the implicit prior works as a black-box from the interpretability point of view. The prior parameters do not represent a clear property of the model in comparison to kernel parameters in standard GPs. Furthermore, even though using 20 samples from the prior has shown to give good results in some cases, there might be situations where this number must be increased, having a big impact in the model's training time. An unexpected result is that the cost of generating continuous samples from a GP prior in DVIP is too expensive. If a GP prior is to be used, it is cheaper to simply use a DGP as the underlying model.

## ACKNOWLEDGMENTS

Authors gratefully acknowledge the use of the facilities of Centro de Computacion Cientifica (CCC) at Universidad Autónoma de Madrid. The authors also acknowledge financial support from Spanish Plan Nacional I+D+i, PID2019-106827GB-I00. Additional support was provided by the national project PID2021-124662OB-I00, funded by MCIN/ AEI /10.13039/501100011033/ and FEDER, "Una manera de hacer Europa", as well as project TED2021-131530B-I00, funded by MCIN/AEI /10.13039/501100011033 and by the European Union NextGenerationEU PRTR.

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
