# OpenReview forum: "Deep Variational Implicit Processes"
_ICLR.cc/2023/Conference — ICLR 2023 poster_

### Official Review · Reviewer_AATZ · 2022-10-18

**Confidence:** 4
**Correctness:** 3
**Technical Novelty And Significance:** 2
**Empirical Novelty And Significance:** 3
**Recommendation:** 6

**Clarity, Quality, Novelty And Reproducibility:**

The paper is well-written, though the notation can be a little dense sometimes. The paper itself is of good quality; the shortcomings of the experiments are really the only negative contributions to the overall quality. With regards to novelty, as mentioned above, I think the paper lacks some significant contribution to be considered novel. The paper appears to be fully reproducible; the authors provided the source code for all experiments, though I did not try to run this.

**Questions for the authors**
- The log-marginal likelihood of the VIP is approximated using the $\alpha$-energy (Eq. (5)), but for DVIP you seem to use the usual variational ELBO. Is there a particular reason for this choice?
- In section 4, you write that DVIPs are expected to have better calibrated uncertainty estimates than VIPs. Why is this?
- The training times for the models on the UCI datasets confuse me a little. From the appendix, it seems you run all experiments for a fixed number of iterations (150k) using a fixed batch size. If by "iteration" you mean "model update", the large differences in training time between the datasets seem puzzling. Does "iteration" instead mean a full pass through the training set, or do you have an idea where the difference may come from?
- The performance of the DGP for the interpolation experiment on the Mauna Loa dataset seems very strange to me. A standard GP would do a much better job here, essentially achieving the same fit as DVIP. Did something go wrong during training, I wonder?

**Strength And Weaknesses:**

### Strengths
The presented model (DVIP) is a very interesting contribution to the field of implicit processes (IPs) and Bayesian deep learning more broadly. It is a very natural extension of shallow IPs, and it is likely to be useful to a significant part of the community, in particular the more applied part. The performance in terms of RMSE and NLL is on par with DGPs, but DVIPs are much, much faster to train. In summary:
- Significant contribution with potentially a high impact on the field.
- Experimental results on par with DGPs, occasionally outperforming them.
- Impressive computational performance.

### Weaknesses
While the contribution is certainly significant due to the impressive experimental results, the novelty appears limited. Despite the authors going through deriving the deep extension of VIPs, they are essentially just using the doubly-stochastic framework derived by Salimbeni & Deisenroth (2017). They can do so since VIPs (which the authors use for each node in the network) are simply GPs, only with an empirically determined mean vector and covariance matrix. This makes VIPs more flexible, but since they are simply GPs, they fit right into the doubly-stochastic DGP (DS-DGP) framework without needing any additional derivations. Thus, the two main components of the model, VIPs and the DS-DGP framework, are already fully developed and constructing the DVIP seems to be just a matter of combining the two. It is a good idea and a natural extension of VIPs, but I do not see any real novelty here. In section 4, the authors write that the DVIP is a generalisation of a DGP, but I disagree with that statement; the DVIP seems to be exactly a DGP. The argument seems to be (last paragraph of section 4) that the DVIP can use IP priors that are not necessarily GPs. While this is true (in the paper, the authors use a BNN as IP prior), the DVIP, as the name suggests, uses VIPs as nodes, which means that the model is still just a DGP; it just uses more flexible GPs as nodes rather than the usual sparse variational GPs.

The experiments are the other major weakness of the paper. While the experiments on the UCI datasets are very thorough, the remaining ones seem somewhat biased towards DVIP. In the interpolation experiment, the DVIP essentially achieves the same fit as an ordinary GP with a very simple covariance function (say, a squared exponential) would; adding such a baseline would be helpful. For the large scale regression experiment and the image classification experiment, only one (or two, in the case of MNIST) DS-DGP architecture is shown, whereas the authors typically show four architectures for the DVIP. Since the best DVIP architecture seems to vary between two and five layers, it doesn't seem fair only to allow the DS-DGP a single column in the tables. Furthermore, for the image classification experiment, the authors use a convolutional layer in their BNN, but they do not use a convolutional kernel (van der Wilk et al. (2017), Dutordoir et al. (2020)) for the DGP. This gives the DVIP an unfair advantage. Lastly, the authors mention that sparse IPs (SIPs) are more expressive than VIPs, being able to model multimodal outputs, though also more computationally demanding. I think a SIP would have made sense to include as an additional benchmark model to see if the deep structure of the DVIP is really necessary.

In summary:
- Limited novelty (builds heavily on the DS-DGP framework).
- The proposed model is essentially a DGP with a more expressive prior.
- Experiments are not entirely fair or transparent. SIP would have been a good baseline to include.

References:
- Salimbeni & Deisenroth (2017): https://arxiv.org/abs/1705.08933
- van der Wilk et al. (2017): https://arxiv.org/abs/1709.01894
- Dutordoir et al. (2020): https://arxiv.org/abs/1902.05888

**Summary Of The Paper:**

The paper proposes a deep extension of variational implicit processes (VIP, Ma et al. (2019)), in which an implicit process, defined as a (nonlinear) transformation of a collection of random variables, is approximated using a Gaussian process. Together with a Gaussian likelihood, this gives a Gaussian process posterior approximation for the implicit process. By extending VIPs to deep architectures, similarly to how deep GPs extend single GPs, the authors show that the resulting deep VIP (DVIP) model outperforms its shallow counterpart and achieves similar performance to a DGP, though with an impressive reduction in training time.

**Summary Of The Review:**

The proposed model (DVIP) is a natural extension of the VIP, which could be a significant and potentially high-impact contribution to the field of Bayesian deep learning. The model shows good experimental performance, and the training time is particularly impressive. However, the model is simply a DGP (albeit more flexible than the traditional DGP) constructed from two well-known parts, VIPs and the doubly-stochastic DGP framework, limiting the novelty significantly. Further to this, the experiments do not always seem fair, which leaves one feeling somewhat sceptical about the claimed superiority of DVIP, which doesn't seem to be significantly better than a DS-DGP in terms of RMSE and NLL. Overall, I feel that the paper falls slightly below the acceptance threshold.

---

### Official Review · Reviewer_cZop · 2022-10-24

**Confidence:** 3
**Correctness:** 4
**Technical Novelty And Significance:** 3
**Empirical Novelty And Significance:** 3
**Recommendation:** 6

**Clarity, Quality, Novelty And Reproducibility:**

Clarity: The paper is generally well-written and clear.

Quality: The paper is technically sound and of good quality.

Novelty: I have raised some concerns about the novelty and contribution of the paper. Please see my comments above.

Reproducibility: Code is provided but I didn't run the code to reproduce the results.

**Strength And Weaknesses:**

Strength:
- The idea of extending the VIP to its deep variant is interesting and in the right direction.
- Experimental results are thorough and encouraging.
- The paper is well-written and easy to follow.

Weaknesses:
- Novelty and contribution: After reading the paper, I found the paper basically extends VIP to its deep version just like the way in the paper DSDGP (Salimbeni & Deisenroth, 2017). The derivations of DVIP are actually very similar to the derivations of DSDGP. It seems that incorporating the VIP priors into the framework of DSDGP is somewhat straightforward. It is unclear what is the main novel technical derivation or novel insights. (Nevertheless, this extension itself is also acceptable to me.)

Questions:
- In Section 4,
    > unlike that of DVIP and the non-linear transformation of DKL ignores epistemic uncertainty, unlike IPs.


     This seems not true. DKL learns an NN-parametrized kernel function while maintaining a GP as the predictive distribution over functions, which represents epistemic uncertainty. In this sense, DKL and DVIP both learn parameters of prior so they can be viewed as empirical Bayesian methods. Have you considered these baselines besides DGPs?

**Summary Of The Paper:**

This paper introduces deep variational implicit processes (DVIP), a deep extension of the prior work "variational implicit processes" (VIP). It can be viewed as a direct generalization of deep Gaussian processes (DGP) by placing implicitly defined multivariate distributions over any finite multivariate marginal of the distribution over functions (e.g., Bayesian neural networks). The fundamental step is to approximate each layer of the implicit process with a GP whose mean and covariance functions are induced by the implicit process. Further linear approximation and reparametrization are adopted for practical training. Finally, the proposed method is validated against several standard benchmarks, with encouraging results.

**Summary Of The Review:**

This paper presents an interesting extension of VIPs with solid technical parts and experimental results. The downside is that the novelty seems to be marginal.

---

### Official Review · Reviewer_Lo8f · 2022-10-24

**Confidence:** 4
**Correctness:** 3
**Technical Novelty And Significance:** 3
**Empirical Novelty And Significance:** 2
**Recommendation:** 6

**Clarity, Quality, Novelty And Reproducibility:**

The paper is clearly written and the quality of the method seems high. The experiments are of somewhat lower quality, although they use a range of different tasks. The novelty is limited since it is a more or less straightforward combination of DGPs and VIPs, but I don't think that should be a problem. The experiments seem reproducible.

**Strength And Weaknesses:**

Strengths:
- Given that DGPs often work better than GPs and VIPs as well, combining them seems like a well-motivated idea.
- The paper is overall clearly written.
- The experiments are quite extensive.

Weaknesses:
- The experiments have mixed results and especially the comparison to the DGP could be done with more care.
- The exact design of the right DVIP model for a given task could be discussed more, also related to prior work.

Major comments:
- How principled is it in the model to fit the prior to the data? Surely, in a proper Bayesian model that wouldn't be allowed. It is said that overfitting is avoided by coupling the priors for different parts of the model, would there be a more principled way, e.g., using hyperpriors?
- There are some related works on functional BNNs that could be mentioned, e.g. [1,2,3]
- There are many works on priors regarding BNNs and stochastic processes, many of which should be mentioned (see [4] for an overview)
- How much does width help in the DVIP model, especially given that the prior functions are the same for all units? Would it be similarly detrimental to the results in [4]?
- I find Fig. 4 quite puzzling because this dataset is often used to show how the periodicity can be modeled. Even a simple GP with linear + periodic kernel can quite reliably do this. From this perspective, all the presented methods seem to miss the main feature of the data, which seems like a strong failure case.
- Error bars would be useful in Tabs. 1 and 2, to judge the significance of the performance differences.
- Contrary to the authors' claim, the DGP seems to yield a *better* likelihood on MNIST than the DVIP with CNN prior. Since the likelihood takes the full posterior into account and is a proper scoring rule, I would lend it more trust than the accuracy. How can the authors explain that the DGP outperforms the DVIP there?
- It is claimed in many experiments where the DGP outperforms the DVIP that the DVIP could be used with GP prior to achieve the same results. I think this should be easy to try and would definitely warrant being included as a baseline. Particularly, I would be interested in the tradeoff between runtime and performance in those cases where both DGP and DVIP use the same GP priors.

Minor comments:
- Sec. 1: is is -> is
- In eq. 6, why are there two products over n? I think there should only be one.
- The notation is often quite laden with sub- and superscripts, maybe that could be simplified.
- How sensible is it to train the model using only one mixture component (effectively a single Gaussian predictive, similar to the normal VIP or GP) and then use 100 components during testing? Would that not result in a stark shift of the kind of distributions that are predicted?

[1] https://arxiv.org/abs/2008.08400

[2] https://hudsonchen.github.io/papers/Tractable_Function_Space_Variational_Inference_in_Bayesian_Neural_Networks.pdf

[3] https://arxiv.org/abs/2106.11642

[4] https://arxiv.org/abs/2105.06868

[5] https://arxiv.org/abs/2106.06529

**Summary Of The Paper:**

The authors propose a Deep Variational Implicit Process (DVIP) which is an extension of the existing VIP model in the same way that a Deep GP (DGP) is an extension of the GP. They show how to perform approximate inference in this model (combining ideas from DGP and VIP inference) and present the empirical performance of the model on a range of experiments.

**Summary Of The Review:**

Overall, I think combining DGPs and VIPs is a nice idea and it is well presented here. Only the experiments could be done with a bit more care and discussed in more nuance, especially regarding the comparison to DGPs and using DVIPs with GP priors as a baseline. If the experiments would be improved, I'd be happy to recommend acceptance. For now, I will lean slightly toward rejection.

UPDATE: Based on the changes made by the authors during the rebuttal, I have updated my score.

---

### Official Review · Reviewer_GNWm · 2022-10-25

**Confidence:** 3
**Correctness:** 3
**Technical Novelty And Significance:** 3
**Empirical Novelty And Significance:** 2
**Recommendation:** 8

**Clarity, Quality, Novelty And Reproducibility:**

For the most part the paper was clearly written (except for things explicitly pointed out above in Strengths and Weaknesses).  The approach is interesting and is well-couched in terms of how it differs from existing work.  I was able to run the code, which is documented impressively well.

**Strength And Weaknesses:**

Strengths:
* The paper presents a comprehensive overview of related work and how the present work fits into this picture.
* The idea of the DVIP itself is clean.  It is presented in a way where it is a natural extension of both deep gaussian processes and variational implicit processes.
* The initial empirical experiments show that the approach could be useful in some settings.

Weaknesses (roughly in order of importance):
* At several points throughout the paper it is noted that DVIP is both a generalization of DGP and also that DVIP is substantially faster than DGP.  In some of the empirical results DGP outperforms DVIP (e.g., HIGGS) and the authors stat that using DVIP with GP priors should give results comparable to (but presumably faster) than DGP.  The authors should consider actually running these experiments (putting GP priors on DVIP) -- a faster method for performing accurate inference for DGPs would be of interest to the community.  If however, the accuracy is not as good, then it suggests that some of the approximations in the DVIP framework result in worse performance (e.g., sampling vs. using inducing points).  In any case, I think that the authors should either back up claims that DVIP can be a faster, but comparably accurate implementation of DGP, or remove such claims from the paper.
* Figure 1 (right) was somewhat confusing to me.  The notation on the bottom suggests that each layer is a function of the inputs (x), but my understanding is that it is a function of the outputs of the previous layer (f^{(t-1)}).
* minor comment: since the prior has hyperparameters that are learned from the data (the means and the variances of the weights in the BNNs), we might expect DVIP to be over-confident.  That is, with enough hyperparameters in the BNNs, DVIP could be overfit to the training data.  The authors mention that the prior mean and variance of the weights in each layer are constrained to be the same, and I suspect that this is to prevent overfitting.  It would be good to be more explicit about this possibility, and to include some results about regimes in which DVIP starts to become too overconfident.
* minor comment: "using 20 samples from the prior has shown to give remarkable results in most cases" seems like too strong of a statement to use here. I do agree that it is interesting (remarkable even) that means and covariances can be estimated with sufficient accuracy using 20 samples, I think the phrasing here should be toned down (for the most part, the results are not _remarkably_ better than other approaches).

**Summary Of The Paper:**

This paper introduces Deep Variational Implicit Processes (DVIPs).  The main idea is to approximate the output of an implicit process (i.e., a random function from which it is easy to drawn samples) as a particularly simple Gaussian Process, and then stack several of these implicit processes with the output of one random function acting as the input to the next.  The authors present a variational inference objective for DVIPs and optimize this objective using sampling-based techniques.  The proposed framework generalizes Deep Gaussian Processes (DGPs) and also Variational Implicit Processes.  The authors apply DVIPs to a number of regression and classification experiments where the perform competitively with existing approaches like VIPs and DGPs.

**Summary Of The Review:**

Overall, I think this paper presents an interesting approach that is for the most part clearly explained, shows promise on some preliminary benchmarks, and has well-documented code.

---

### Decision · Program_Chairs · 2023-01-20

**Decision:**

Accept: poster

**Justification For Why Not Higher Score:**

In light of the concerns about novelty, the contributions probably don't reach the level of significance for a spotlight.

**Justification For Why Not Lower Score:**

This paper seems like a useful contribution, and the reviewers don't seem to have identified any significant flaws that would merit rejection.

**Metareview: Summary, Strengths And Weaknesses:**

This paper presents a deep extension of variational implicit processes, in the same way that deep GPs are a deep extension of GPs. It builds on the doubly stochastic GP framework of Salimbeni and Deisenroth (2017), but uses inference algorithms based on explicit function samples (as required for VIP). The reviews were originally borderline; reviewers appreciated the clean presentation and the detailed discussion of the literature, as well as the high-quality code base provided. Multiple reviewers had concerns about the limited novelty and about the fairness of comparisons against GPs. The authors provided extensive new experiments in the revision which led the reviewers to raise their scores, and now all reviewers are in favor of acceptance. Regarding novelty, it is true that the overall idea is a combination of two existing ideas. However, it's not a matter of just clicking pieces together; reviewers feel that the algorithmic details are interesting and well thought out. Overall, I recommend acceptance.


**Note From Pc:**

if the above contains the word "oral" or "spotlight" please see: "oral" presentation means -> notable-top-5% and "spotlight" means -> notable-top-25%. As stated in our emails, we are disassociating presentation type from AC recommendations